# Cost-effectiveness of hypertension therapy based on 2020 International Society of Hypertension guidelines in Ethiopia from a societal perspective

Majid Davari[1☯], Mende Mensa Sorato [1,2☯]*, Abbas Kebriaeezadeh[1‡], Nizal Sarrafzadegan[3,4‡]

1 Department of Pharmacoeconomics and Pharmaceutical Administration, The Institute of Pharmaceutical Sciences, Tehran University of Medical Sciences, Tehran, Iran, 2 Department of Pharmacy, College of Medicine and Health Sciences, Arba Minch University, Arba Minch, Ethiopia, 3 Isfahan Cardiovascular Research Center, WHO Collaborating Center, Cardiovascular Research Institute, Isfahan University of Medical Sciences, Isfahan, Iran, 4 School of Population and Public Health, University of British Columbia, Vancouver, Canada

☯ These authors contributed equally to this work.
‡ AK and NS also contributed equally to this work.
* mendemensa@gmail.com

**Data Availability Statement:** All the data reported in the manuscript are available on Supplementary files entitled "Markov model data for CEA hypertension" published with this manuscript.

## Abstract

### Introduction

There is inadequate information on the cost-effectiveness of hypertension based on evidence-based guidelines. Therefore, this study was conducted to evaluate the cost-effectiveness of hypertension treatment based on 2020 International Society of Hypertension (ISH) guidelines from a societal perspective.

### Methods

We developed a state-transition Markov model based on the cardiovascular disease policy model adapted to the Sub-Saharan African perspective to simulate costs of treated and untreated hypertension and disability-adjusted life-years (DALYs) averted by treating previously untreated adults above 30 years from a societal perspective for a lifetime.

### Results

The full implementation of the ISH 2020 hypertension guidelines can prevent approximately 22,348.66 total productive life-year losses annually. The incremental net monetary benefit of treating hypertension based was $128,520,077.61 US by considering a willingness-to-pay threshold of $50,000 US per DALY averted. The incremental cost-effectiveness ratio (ICER) of treating hypertension when compared with null was $1,125.44 US per DALY averted. Treating hypertension among adults aged 40–64 years was very cost-effective 625.27 USD per DALY averted. Treating hypertensive adults aged 40–64 years with diabetes and CKD is very cost-effective in both women and men (i.e., 559.48 USD and 905.40 USD/DALY averted respectively).

**Funding:** The authors received no specific funding for this work.

**Competing interests:** The authors declare that they have no competing interests.

**Abbreviations:** ACEIs, Angiotensin-Converting Enzyme Inhibitors; BP, Blood Pressure; CVD, Cardiovascular Diseases; DALY, Disability Adjusted Life Years; DBP, Diastolic Blood Pressure; EDHS, Ethiopia Demographic Health Survey; HDL, High-Density Lipoprotein; ICER, Incremental Cost-Effectiveness Analysis; LDL, Low-Density Lipoprotein; LMICs, Low- and Middle-income Countries; MI, Myocardial Infarction; NCDs, Non-Communicable Diseases; NSAIDs, Non-steroidal Anti-inflammatory Drugs; OSA, Obstructive Sleep Apnea; PACK, Practical Approach to Care Kit; QALY, Quality Adjusted Life Years; SA, Stable Angina; SBP, Systolic Blood Pressure; TIA, Transient Ischemic Attack; UA, Unstable angina; VLDL, Very Low-Density Lipoprotein; WHO, World Health Organization; YLD, Years Lived with Disability; YLL, Years of Life Lost.

## Conclusion

The implementation of the ISH 2020 guidelines among hypertensive adults in Southern Ethiopia could result in $9,574,118.47 US economic savings. Controlling hypertension in all patients with or with diabetes and or CKD could be effective and cost-saving. Therefore, improving treatment coverage, blood pressure control rate, and adherence to treatment by involving all relevant stakeholders is critical to saving scarce health resources.

## 1. Introduction

Hypertension remains the leading cause of death globally, accounting for 10.4 million deaths per year [1]. It is associated with societal and economic consequences particularly in Low and middle-income countries (LMICs). In addition to the direct costs associated with health care utilization for the management of complications, hypertension causes significant productivity loss from disability and premature death [2, 3]. According to a global health estimate in 2016, the productive healthy life year lost due to hypertension in the African region was estimated to be 19,395,946 [4, 5].

Treating hypertension is associated with improved life expectancy, despite the disutility penalty associated with daily use of anti-hypertensives [6]. For example, a modeling study based on primary prevention trials showed that for patients with an initial BP of 160/95mmHg, those with antihypertensive treatment and diabetes, or antihypertensive treatment, diabetes, and currently smoking had corresponding gains of life expectancy of 12.50, 2.52, and 2.45 years, with a reduction in BP to 140/82mmHg [7]. The Cost-effectiveness of hypertension treatment is influenced by both the absolute initial cardiovascular risk, and the relative risk reduction [8]. The cardiovascular disease policy model is used by many hypertension cost-effectiveness studies. It is a state-transition (Markov cohort) model of coronary heart disease and stroke incidence, prevalence, mortality, and costs over the age of 35 years [9–11].

Hypertension treatment is influenced by several factors including patient demographics like age, sex, race, and availability of evidence-based guidelines. In addition to this, cost-effectiveness strongly depends on social determinants. Hence, results obtained in one country may not be valid in another [12]. Therefore, it is important to know the value of money invested in treating hypertension, since controlling hypertension could not restore the ideal cardiovascular and cerebrovascular risks to that of the non-hypertensive population. In addition to this disutility associated with daily use of anti-hypertensive medicines mandates who should be treated and who should not. There was no study concerning the cost-effectiveness of treating hypertension patients based on evidence-based guidelines. To fill this evidence gap, generalized cost-effectiveness analyses (GCEAs) from a societal perspective were conducted in Southern Ethiopia by using a modified cardiovascular disease policy model [13–17].

## 2. Methods and materials

### 2.1. Study area and period

The study was conducted among three selected Hospitals in Southern Ethiopia namely Gamo, Gofa, and South Omo Zones starting from September 1, 2020 –to November 30, 2020, to generate direct medical and non-medical cost information. Southern nations and nationalities and Peoples Region (SNNPR) is one of the largest regions in Ethiopia, accounting for more than 10% of the country's land area and an estimated population of 20,768,000 (May 2018)

almost a fifth of the country's population. The SNNPR region is divided into 12 administrative zones [18, 19]. Three Hospitals (Arba Minch, Sawula, and Jinka General Hospital) with experience in providing CVD care for five years or more in selected zones were included.

## 2.2. Study design

The cardiovascular disease (CVD) policy model is a computer-simulation, state-transition (Markov cohort) model of coronary heart disease and stroke incidence, prevalence, mortality, and costs in the population over age 35 years [9–11]. The Model was used by different countries to evaluate the cost-effectiveness of hypertension treatment including the USA and China [16, 20]. A modified CVD policy model for Sub-Saharan Africa was used to simulate hypertension management, drug treatment, disease-related costs, and disability-adjusted life years (DALYs) averted by preventing CVD or loss in untreated hypertensive adults aged > 30 years over a lifetime horizon [15]. We run the model for 80 cycles with a cycle length of one year. The cost-effectiveness of treating hypertension-based ISH guidelines when compared with no treatment was conducted from a societal perspective. All methods were performed under relevant guidelines.

## 2.3. Population

The study populations are selected patients and their follow-up records of adult hypertensive patients in four selected Hospitals.

## 2.4. Eligibility criteria

All adult hypertensive patients having at least five years of follow-up visits before data collection and receiving care during the study period from selected facilities were included. However, patients who are unwilling to participate in this study, and patients who have less than five years of follow-up were excluded.

## 2.5. Study variables

### 2.5.1. Dependent variables.

- Cost-effectiveness of hypertension treatment based on 2020 ISH guidelines.

### 2.5.2. Independent variables.

- Patient-related (socio-demographic characteristics, heart disease knowledge, healthy lifestyle and heart disease risk perception, presence of comorbidity, type of medications, treatment adherence, shared decision making, health-related quality of Life)

- Health professional related (sociodemographic characteristics, qualification, experience, knowledge of evidence-based guidelines)

### 2.5.3. Cost related variables.

○ **Medical costs** (inpatient hospital stay/hospitalization cost, outpatient clinic visit, drug acquisition costs, drug administration cost, laboratory test, and imaging study costs)

○ **Non-medical costs** (transportation, meal, patient time cost due to treatment, cost due informal care by family or friends)

○ **Indirect costs** (absenteeism, presenteeism, unemployment, early retirement, disability, premature death)

## 2.6. Sample size and sampling technique

**2.6.1. Sample size determination.** The sample size was determined by using the single population proportion formula by taking the prevalence of patients who controlled their BP as 14% from the WHO 2016 BP control rate report [21–23] and Z value of 1.96 at a 95% confidence interval. We added 10% for non-response rate and two for design effect due to multistage sampling technique involvement. Finally, a formula giving a larger sample size was used. A total of 407 hypertensive adult patients who are on follow-up care will be included.

$$n = \frac{(Za/2)^2 \ P(1-P) = 185}{d^2}$$

$$= 185 + (185 * 10\%) = 203.5$$

$$= 203.5 * 2 = 407$$

**Where: n** = is the sample size

- $Z^2$ = standard normal deviation, set at 1.96, correspond to the 95% confidence interval

- **d** = is the desired level of precision/margin of error (0.05)

- **p** = prevalence of patients taking anti-hypertensive (p=28.4%), and q is 1-p.

**2.6.2. Sampling techniques.** A multi-stage simple random sampling technique was used. We randomly selected three zones from a total of 12 zones found in the Southern region. Three general hospitals with experience in providing CVD care for at least five years from selected three zones were included in this study. The total sample size was allocated to these hospitals based on an estimated number of adult hypertensive patients attending respective hospitals (i.e., we included 212 patients from Arba Minch General hospital, 107 patients from Jinka general hospital, and 88 patients from Sawula general hospital). Finally, a consecutive sampling technique was applied in each facility until the desired sample size was achieved.

## 2.7. Data collection tools and procedures

Key model inputs variables include; the 2020 population of selected Zones, hypertension prevalence by treatment and control status, transition probabilities to death and healthy state, cost of diagnosis, and management. The data was collected from the National STEPS survey [24], systematic reviews [25–28], and our effectiveness study. Treated and controlled hypertension was defined based on the BP control target of ISH 2020 guideline (i.e., controlled, if BP < 130/80 mmHg for < 65 years and < 140/90 mmHg for ≥ 65 years and uncontrolled, if BP ≥ 130/80 mmHg for < 65 years and ≥ 140/90 mmHg for ≥ 65 years) [1].

**2.7.1. Disease states and transition probabilities.** The 2020 world population prospect estimate was used for the baseline population [29]. Coronary heart disease and stroke deaths in 2020 were extracted from the WHO-STEPS survey and systematic reviews. Coronary heart disease deaths and stroke deaths were CVD deaths and all other deaths were considered non-CVD deaths. The annual probability of CHD and stroke, data that are not available can be taken from well-accepted international studies like Framingham Heart Study due to lack of country-specific [30] and the Framingham Offspring Study [31], by contextualizing to Ethiopian scenario. The Framingham offspring studies 1–8 presented the 10-year CHD and stroke events, and we used annual CHD and stroke risk. We tested our annual risk assumption, by

developing a scenario analysis with a linearly increasing risk of CVD events over the 10 years [32].

The mortality rate in 2015 stratified by sex and 5-year age groups in selected zones was used. We combined disease-related and disease-unrelated mortality in a multiplicative fashion. The demographic profile of the cohort was derived from the estimated resident population of Ethiopia in 2020. The rate of BP control was drawn from the National STEPs 2015 survey [24] and our effectiveness study. Identified rates were applied to projections from the 2019 United Nations World Population Prospects [33]. Transition probabilities (TP) and relative risk of mortality were taken from the natural history of hypertension studies with good quality [34–39] (S1 and S2 Tables).

The prevalence of cardiovascular risk factors (MI, angina, heart failure, stroke, TIA) was estimated from systematic reviews and meta-analyses. Incorporating the risk factor prevalence data in the relevant Framingham risk equation, the age and sex-specific probability of CHD and cerebrovascular disease (i.e., stroke and TIA) events were estimated. The probability of each health state was calculated using the age- and sex-specific CHD and cerebrovascular disease event distributions [24, 40]. To estimate the corresponding probabilities, separate relative risk estimates were used for CHD events (Angina, HF, and MI) and cerebrovascular diseases (stroke and transient ischemic attack). We assumed antihypertensive treatment affects the probability of every disease state similarly across all age and sex groups. Relative risk reductions attributable to antihypertensive treatment were extracted from the peer-reviewed literature [39, 41, 42]. We estimated the probability of death separately for (1) all-cause mortality in absence of hypertension and related complications for the productive age population and (2) mortality attributable to the included disease states. The first component was estimated using Ethiopian Life Table 2019, and the second component was calculated based on standardized mortality ratios extracted from literature [34] (S2 and S3 Tables).

Interventional trials suggested that it could be possible to achieve effective BP targets in about 70% of patients by improving adherence and/or intensifying therapy [35]. Incident coronary heart disease events were allocated to angina pectoris, MI, or cardiac arrest. Prevalence and joint distributions of Ethiopia's risk factor values were estimated from the national STEPS survey [24]. Betas for risk function for non-blood pressure risk factors were estimated separately for the risk of incident coronary heart disease events, incident strokes, and non-CVD deaths from the Framingham offspring cohort [31]. Risk factors are assumed to affect the incidence of MI, arrest, and angina in proportion to the overall incidence of CHD, except tobacco smokers are assumed to have a higher relative risk for infarction and arrest [43]; and a proportionately lower coefficient for angina [24, 44]. Case-fatality rates, rates of MI, prehospital arrest deaths, and out-of-hospital cardiac arrests surviving to hospital discharge were estimated from evidence-based literature (S2 and S3 Tables).

Survival after a CHD event was estimated from international data sources (California data on the ratio of in-hospital survival to 30-day survival) [45] and calibrated based on the findings of Huffman et al. [46]. Stroke incidence was assumed to be independent of the risk of new-onset coronary heart disease in the same year. The number of hospitalized stroke cases was obtained from national and regional studies. The annual probabilities of stroke after MI [47, 48] and the probability of CHD in stroke patients were based on natural history studies and systematic reviews of BP control trials [49–54]. A 30-day heart failure mortality and re-hospitalization data were from the THESEUS-HF registry [55] and Korean Acute Heart Failure Registry (KorAHF) [56, 57] (S2 and S3 Tables). The background prevalence of CVD by age, sex, and CVD disease state (stroke, coronary heart disease, or both stroke and coronary heart disease) in 2020 was estimated from GBD 2017 [58] (S2 Table).

**2.7.2. Cost estimation.** Both direct and indirect costs were included in this study. The direct costs were divided into two subcategories: direct medical costs and direct non-medical costs. Direct medical costs include; inpatient stays, outpatient clinic visits, medical services, drug acquisition, dispensing, administration, monitoring, laboratory test, and imaging study costs. The costs associated with outpatient/inpatient visits were estimated by multiplying the numbers of outpatient visits related to hypertension by the outpatient costs per year (i.e., twelve times WHO cost per outpatient visit for secondary hospitals inflated to 2021) [59].

Data concerning medications prescribed for the management of hypertension, and associated comorbidities, and laboratory tests and imaging studies were collected by patient chart abstraction in index year (2020). The cost of medications used for management of hypertension and associated comorbidities was taken from Ethiopian Pharmaceutical supply agency Arba Minch regional hub selling price and retail price of Arba Minch General Hospital in 2020. The retail price of Arba Minch General Hospital was used because of the minimum distance from the Pharmaceutical supply agency hub, which could minimize the markup added to the retail price due to transportation costs. Costs of laboratory tests and imaging studies were also taken from Arba Minch Hospital Laboratory's service price list. The salary scale of the health workforce was based on the federal ministry of health (FMOH) of Ethiopia (S4 Table).

Ongoing program costs for hypertension care were estimated from WHO tool outputs for CVD and diabetes care and the National strategic action plan (NSAP) for prevention & control of non-communicable diseases in Ethiopia from 2014-to 2016 and adjusted for the 2021 inflation target population [60]. Adjustment for the study population was done by multiplying the national cost by the proportion of the study population (i.e., 3%). National and regional cost estimates were based on the proportion of patients studied (i.e. 3% and 20%). We considered this strategy since the age and sex distribution of hypertension among different regions in the country did not vary significantly. The collected cost data were added up and averaged by using a bottom-up approach. Facility-based or reference costs were used during computing costs. The total medical cost of hypertension treatment was calculated as the sum of the product of medical costs with their respective unit prices. Costs were discounted at an annual rate of 3% and reported in 2021 USD [13, 61].

Direct non-medical costs include transportation costs and patient time costs due to care. The cost of patient time due to care was estimated by using the average daily wage of the patient (97.00 WTB) which was calculated from the average monthly income (2910.00 ETB) from our treatment effectiveness survey. Transportation cost was determined by using the cost of average traveling distance and local transportation tariff (42.00 ETB) in January 2021. According to EDHS 2016 survey showed that 33% of women and 88% of men are currently employed [34]. This proportion was used to determine the patient time cost due to care for employed groups. For the unemployed proportion, the average daily wage of daily laborers workers working 8 hours per day for 6 days per week was used (26.53 ETB) from the monthly wage of 796.00 ETB (420–1172 ETB) [62].

Indirect costs include productivity loss due to illness and the cost of death. Cost-of hypertension-related hospitalization was taken from WHO Choice [59], costs per inpatient stay and cost per inpatient bed day times duration of hospitalization inflated for 2021, and professional time (physician, nurse laboratory professional, and pharmacist time). If a patient had multiple admissions during the year, the costs for each admission were aggregated as the total costs [63]. Age and sex-specific mortality rates among the adult general population in Ethiopia were taken from EDHS 2016 survey and extrapolated to selected populations [34]. According to EDHS 2016, the probability of dying before age 50 years among adults $\geq$ 15 years was 10% and 12%, in women and men respectively [34]. Mortality risk in the general population attributed

to those with and without hypertension using sex-specific estimates of the relative risk (RR) of all-cause mortality associated with hypertension by treatment and control status was derived from a study conducted in India [64]. A cohort study conducted in India among adults 20 years and above to determine the rate and risk of all-cause mortality among people with HTN showed that the incidence of deaths in the study was 4.28%. The relative risk of mortality was 3.13 (CI: 2.91–3.37) and 1.2 in the high BP group and at age of 60 years. The age-adjusted hazard ratio of all-cause mortality for the high BP group was 2.96 (2.56–3.42) [64] (S6–S9 Tables).

In 2020 crude death rate of the Ethiopian population-based on global estimates was 6.29 deaths per 1000 population (i.e. 680,032 deaths per 108,113,150) [65]. The estimated prevalence of hypertension among adults was calculated from National STEPS Survey 2016, systematic review and meta-analysis, World health organization report, and local studies (19.6%, for 15–30 years, 23% for 30–40 years, 25.9% for 40–49 years and 41.9% for 50 years and above [24, 34, 64, 66–69]). (S3 Table). The mean estimated prevalence of hypertension is 21.39%. The mean relative risk (hazard ratio) of all-cause mortality among the hypertensive population when compared to those without hypertension was 1.39 (0.95 to 1.95) [70]. Only 28.4% of patients with the diagnosis of hypertension were taking antihypertensive medication [24].

Years of life lost due to hypertension were determined by first calculating disability weights for specific ages based on BP control status (X). Then subtract this value (X) from the life expectancy of the Ethiopian population (i.e., 66.7 years for men, and 70.4 years for women) (Y). The productivity loss cost due to hypertension was calculated by multiplying Y with the sex-specific employment rate based on a monthly average income of 2059.078 ETB from the National STEPS survey 2015 adjusted for 2021 inflation (1.372) STEPS Survey, 2015 [24] and EDHS 2016 survey showed that 33% of women and 88% of men are currently employed [34] and for unemployed, 2019 minimum average monthly earnings (ETB) of daily laborers reported by the Ethiopian Ministry of Labor and Social Affairs (MOLSA) 796 ETB (420–1172 ETB) [62]. Concerning, the cost of productivity lost due to premature mortality: first we calculated potential years of life lost (YLL) by subtracting life expectancy from the sex-specific age of death at which the death is recorded (Z). Then Z is multiplied by the number of deaths in each age group (Xi). Finally, we multiplied Xi with sex-specific employment rates like productivity loss due to hypertension-related morbidity above [71]. Excess mortality and morbidity due to hypertension to hypertension were determined by subtracting age and sex-specific morbidity and mortality among the general population from the hypertensive cohort. Both were determined by using age, sex, and blood pressure treatment status mortality rate per 1000 person-years (S9 Table).

**2.7.3. Calculating Disability Adjusted Life Years (DALYs).**   Disability-adjusted life years (DALYs) are composed of years lived with disability (YLDs) and years of life lost due to premature mortality (YLLs) [72]. The YLDs arising from a disease or injury are the sum of the YLDs for each of the sequelae associated with that disease. The GBD 2015 study estimated YLDs by country, age, and sex for 2619 sequelae of 310 diseases and injuries [72]. The YLDs (morbidity component) of the DALYs, are calculated as follows [4]:

$$YLD = \textit{Number of cases x duration till remission or death x disability weight}$$

Disability weights of hypertensive heart disease, complications, and comorbidities were taken from WHO estimates of global health state weights, systematic reviews, and clinical or observational studies [73]. According to the Global burden of disease 2016 report, the average disability weight of hypertensive heart disease was 0.246 (0.201–0.300). Untreated 0.323 and treated and controlled 0.171 [74, 75] (S2 Table).

Years lost to live (YLLs), the mortality component of the DALYs, is calculated as follows:

$$YLL = Number\ of\ deaths\ x\ Life\ expectancy\ at\ the\ age\ of\ death$$

DALYs for a specific cause are calculated as the sum of the YLLs from that cause and the YLDs for people living in states of less than good health resulting from the specific cause:

$$\mathbf{DALY}_{(c,s,a,t)} = \mathbf{YLL}_{(c,s,a,t)} + \mathbf{YLD}_{(c,s,a,t)}$$

- For a given cause c, age a, sex s, and year t

According to the GBD, 2010 study simplified calculation of DALYs was recommended. Therefore, we used a new normative standard life table for the loss function used to compute YLLs; calculate of YLDs simply as the prevalence of each sequela multiplied by the relevant disability weight; adjust for comorbidity in the calculation of YLDs, and no discounting for time or unequal age weights [76].

**2.7.4. Adjusting for comorbidity.**   Patients with hypertension may have more than one disease, the addition of YLDs across causes may result in an overestimation of the total loss of health [77]. Therefore, it is recommended to estimate comorbidities using the assumption of independence within age-sex groups [4]:

$$\mathbf{P_{1+2}} = \mathbf{P_1} + \mathbf{P_2} - (\mathbf{P_1\ x\ P_2}) = \mathbf{1 - (1 - P_1)\ x\ (1 - P_2)}$$

**Where** $P_{1+2}$ is the prevalence of the two comorbid diseases 1 and 2,
$P_1$ is the prevalence of disease 1 and $\mathbf{P_2}$ is the prevalence of disease 2.
The combined disability weight for individuals with multiple conditions is estimated assuming a multiplicative model as follows:

$$\mathbf{DW_{1+2}} = \mathbf{1 - (1 - DW_1)\ x\ (1 - DW_2)}$$

Since prevalence YLDs are calculated for each cause as:

$$\mathbf{YLD_i} = \mathbf{DW_i\ x\ P_i}$$

Two preceding equations can be combined into a single calculation resulting in:

$$\mathbf{YLD_{1+2}} = \mathbf{1 - (1 - YLD_1)\ x\ (1 - YLD_2)}$$

**2.7.5. Cost-effectiveness of hypertension treatment.**   Generalized CEA was conducted from a societal perspective to evaluate the cost-effectiveness of hypertension treatment based on the ISH 2020 guideline [78]. Generalized cost-effectiveness analysis (GCEA) was used to derive average cost-effectiveness ratios (ACERs) for the full implementation of ISH guidelines. The Cost-effectiveness of interventions is judged by their incremental cost-effectiveness ratio (ICER) which is given as the ratio of the incremental cost of the intervention to its incremental health gain relative to a comparator. The ratio was reported as cost expressed in monetary units per health gain (e.g., the cost in USD per disability-adjusted life year (DALY) averted). ICER informs us of how much additional cost the intervention under consideration requires for a unit increase in health benefits over its comparator. Therefore, the lower the ratio, the more cost-effective the intervention is [13]. ICERs equal to one to three times the per capita gross domestic product of Ethiopia (936 USD in 2020) [79] were considered cost-effective. An ICERs below USD 936.30 per DALY averted was considered very cost-effective and above 2,808.90 USD (122,187.15 ETB) were not cost-effective. In addition to this, we used half of the

GDP per capita threshold (468.15 USD or 20,364.53 ETB) as it is more in accord with countries' realities than the 1–3 GDP per capita, and could be used as an interim rule of thumb rather than the 1–3 GDP rule [80]. Results were presented based on Consolidated Health Economic Evaluation Reporting Standards (CHEERS) [81].

**2.7.6. Model assumptions and transition probabilities.** When the hypertensive population enters the model they can be in one of two health states: a treated and untreated hypertension. People in a treated/untreated state move to a treated controlled/uncontrolled state, then people in a controlled or uncontrolled state can move to a non-fatal CHD and stroke event state or the dead state. Once people are diagnosed as having hypertension, they receive antihypertensive treatment, and they will get a reduction in their risk of having a cardiovascular event [82]. We assumed that truly normotensive people can become hypertensive over time because there is an additional possibility in a cycle that they will have a BP check-up annually and move back to a hypertension state. The probability will be estimated from the annual probability of developing hypertension based on age and sex distribution. Once a person has moved to a non-fatal event state (MI, UA, SA, stroke, or TIA) they stay there unless they die. This was considered a reasonable simplification for modeling purposes and also reflected in 12 trials where an individual is censored at their first event. The non-fatal CHD and stroke events were assumed to have two states; event and post-event. This is so that a different cost can be applied in the first cycle reflecting acute management and/or diagnostic costs. The event state is a tunnel state where patients move automatically to the post-event state in the next cycle (unless they die). The probability of death due to the CVD states will vary by type of event. Once people have moved to the dead state in the model they cannot of course move elsewhere; this is known as an absorbing state. To avoid modeling errors, we used the TECH-VER verification checklist containing five domains [input calculations; event-state calculations; result calculations; uncertainty analysis calculations; and other overall checks (e.g. validity or interface)] to reduce errors in models and improve their Credibility [83].

**2.7.7. Blood pressure treatment assumptions.** According to guidelines, after a brief trial period of lifestyle interventions, patients with hypertension selected for pharmacologic treatment were grouped into those with or without diabetes or CKD. For the initial choice of antihypertensive agent, blacks were recommended to start with any thiazide diuretic, calcium channel blocker (CCB) not necessarily ACEIs/ARBs. Patients with diabetes and CKD were recommended to start with an ACE or an ARB. The BP response to different agents is similar in most patients [84] and we chose not to model differential use of antihypertensive medication classes in order not to bias cost-of-treatment inputs. Antihypertensive dose intensification and frequency of BP monitoring were based on the heart's technical package guideline for blood pressure control. We did not simulate the effects of any particular medication; instead, we simulated "standard dose" effects and assumed average drug prices across classes [85] (S10 and S11 Tables).

The amount of BP change was assumed to be a function of the baseline BP and the effect of a standard-dose antihypertensive agent at that pre-treatment level [86] (S9 and S10 Tables). It is important to note that for patients with very high BP (mean SBP of $\geq 185$ mmHg) it is assumed that even with taking four standard dose medications, these patients would on average achieve a BP of about 143 mmHg, but not a target of <140 mmHg. Another important consideration is the effect of true treatment-resistant hypertension (TRH) [87]. We did not include the effect of TRH on costs and outcomes due to a lack of national data. Other important assumptions include, that hypertensive deaths reported or recorded on hospital registries were considered hypertension-related unless specified as due to other causes. All Disabilities in hypertensive patients are considered as disabilities due to hypertension except for

comorbidities and accidents. Both costs and outcomes for cost-effectiveness analysis were discounted at 3 and 6% discount rates.

## 2.8. Data quality control, processing, and analysis

Questionnaires are prepared in English and the patient interview part of the questionnaire was translated into Amharic and translated back into English to check its consistency. The Amharic version of the patient interview questionnaire and English version of the health professional interview, data abstraction form, and health system interview questionnaires was used for data collection. The questionnaire was pretested on 30 adult hypertensive patients in Arba Minch General Hospital to ensure that the respondents could understand the questions and to check for consistency and possible amendments were made based on findings. Six professional nurses (BSc.) for data collection and one senior professional working in the respective health facilities for supervision were oriented before data collection about data collection approaches and contents of the data collection format for one day by the principal investigator. Continuous follow-up and supervision were made by the principal investigator throughout the data collection period.

The collected data were checked for completeness and consistency by the principal investigator on daily basis on the spot during the data collection time. Then data were transcribed back to English for the patient interview part and entry was made using Epi-data 3.1 software. After data processing, analysis was done by using SPSS version 21.0 and Microsoft excel 2010. A summary of descriptive statistics was computed for productivity loss; direct and indirect costs, and the cost-effectiveness of hypertension treatment, and the results were presented in figures and tables.

## 2.9. Ethics approval and consent to participate

The study was approved by Tehran University of medical sciences, Faculty of pharmacy, department of pharmacoeconomics, and pharmaceutical administration ethical review board with Approval ID: *IR.TUMS.MEDICINE.REC.1399.674* and Arba Minch University College of medicine and health sciences Institutional review board with Reference number: *IRB/T10/2012*. After clarifying the study objective and confidentiality of the information; verbal informed consent was obtained from each respective hospital before data collection. All methods were performed under relevant guidelines.

## 3. Results

### 3.1. Description of the study

We used a state-transition Markov model for the population over the age of 30-years based on the cardiovascular disease (CVD) policy model adapted for the Sub-Saharan African context [9–11, 15]. Basic inputs used during model development and source of information were described below (Table 1). Concerning treatment modalities, six antihypertensive treatment packages (17 modalities), and six primary and secondary prevention and treatment packages were included (Table 2). Based on the ISH 2020 guidelines, approximately 16,661,325 adults aged 15 years and above were with high blood pressure in Ethiopia. Out of this only 28.4% (4,731,816) were taking antihypertensive medications and the remaining 71.6% (11,929,509) were not taking antihypertensive medications and eligible for treatment [24, 34, 64, 66–69]. In our study, treating hypertension to ISH 2020 guidelines target was projected to prevent approximately 22,348.66 total productive life-year losses annually. This equates 9,574,118.47 $US in 2021 (Table 3).

**Table 1. Variables and input data for evaluating the cost-effectiveness of hypertension treatment based on 2020 ISH guidelines in Southern Ethiopia, January 2021.**

| List of medicines | Unit | Retail Price in 2021 USD | Source |
|---|---|---|---|
| Acetylsalicylic Acid - 81mg–Tablet (Enteric Coated) | 10x10 | 1.303 | Ethiopian Pharmaceutical supply agency, Arba Minch Hub wholesale price 2021 and Arba Minch General hospital pharmacy retail price 2021 |
| Adrenaline (Epinephrine)-0.1% in 1mL ampoule | Each | 1.074 | |
| Amiodarone - 100mg–Tablet | 10x3 | 9.337 | |
| Amlodipine - 10mg—Tablet | 10x10 | 3.142 | |
| Amlodipine - 5mg–Tablet | 10x10 | 2.243 | |
| Atenolol - 50mg–Tablet | 10x10 | 1.749 | |
| Atorvastatin - 20mg–Tablet | 10x10 | 5.831 | |
| Atorvastatin - 40mg–Tablet | 10x3 | 4.195 | |
| Beclomethasone Propionate -100mcg/dose–Aerosol (Oral Inhalation) | 200 MD | 3.929 | |
| Candesartan - 8mg–Tablet | 14x2 | 4.548 | |
| Captopril—12.5mg–Tablet | 10x10 | 1.000 | |
| Captopril - 25mg–Tablet | 10x10 | 0.802 | |
| Dexamethasone - 4mg/ml in 1ml Ampoule—Injection | 10 | 0.118 | |
| Captopril + HCT (50mg + 25mg)-Tablet | 10x10 | 1.708 | |
| Digoxin—0.25mg–Tablet | 10x10 | 6.025 | |
| Enalapril Maleate - 10mg—Tablet | 10x10 | 1.835 | |
| Enalapril Maleate - 5mg–Tablet | 10x10 | 1.905 | |
| Enalapril Maleate– 2.5mg–Tablet | 10x10 | 0.595 | |
| Enalapril Maleate +HCT (10 mg + 25 mg)-tablet | 10x10 | 2.331 | |
| Glibenclamide - 5mg–Tablet | 10x10 | 1.165 | |
| Glucose 40% in 20 mL–IV infusion | Each | 0.076 | |
| Glyceryl Trinitrate—0.4mg–Tablet (Sublingual) | 100 | 14.518 | |
| Hydralazine - 20mg/ml in 1ml ampoule—Injection | 5 | 6.079 | |
| Hydrochlorothiazide - 25mg–Tablet | 25x4 | 1.432 | |
| Insulin Isophane Biphasic (Soluble/Isophane Mixture)- (30 + 70) IU/ml in 10ml Vial -Injection (Suspension) | Each | 2.539 | |
| Insulin Isophane Human - 100IU/ml in 10ml Vial -Injection (Suspension) | Each | 2.988 | |
| Insulin Soluble Human - 100IU/ml in 10ml Vial | Each | 3.165 | |
| Lovastatin - 20mg–Tablet | 10x10 | 2.521 | |
| Metformin - 500mg–Tablet | 10 | 0.828 | |
| Methyldopa - 250mg–Tablet | 100x10 | 1.542 | |
| Metoprolol - 50mg–Tablet | 10x10 | 2.814 | |
| Morphine sulphate-30mg-tablet | 110 | 12.239 | |
| Nifedipine - 20mg–Tablet | 10x10 | 1.749 | |
| Prednisolone—5 mg–Tablet | 100x10 | 10.198 | |
| Propranolol - 40mg–Tablet | 10x10 | 2.013 | |
| Propylthiouracil - 100mg—Tablet (Scored) | 100 | 18.889 | |
| Salbutamol—0.1mg/dose—Aerosol (Oral Inhalation) | 200 MD | 3.492 | |
| Spironolactone - 25mg–Tablet | 10x10 | 2.440 | |
| Thyroxin Sodium—0.1mg–Tablet | 100 | 5.319 | |
| Valsartan + HCT (80mg +12.5mg) | 7*2 | 1.146 | |
| **Laboratory and imaging costs** | | | |

(*Continued*)

**Table 1.** (Continued)

| List of medicines | Unit | Retail Price in 2021 USD | Source |
|---|---|---|---|
| Complete blood count | | 1.72 | Arba Minch General Hospital Laboratory service price 2021 |
| Fasting/random blood sugar | | 0.46 | |
| Lipid profile (LDL, HDL, Total cholesterol, Triglyceride) | | 3.68 | |
| Echocardiography | | 2.76 | |
| Electro cardiogram | | 8.05 | |
| Computed Tomography scan | | 27.59 | |
| Renal function test (bilirubin, creatinine) | | 1.84 | |
| Chest X-ray | | 16.69 | |
| Urine analysis | | 0.34 | |
| Body fluid analysis | | 2.30 | |
| H.pylori | | 1.15 | |
| Liver function test (AST, ALT, ALP) | | 2.76 | |
| Thyroid function test (T3, T4, TSH) | | 9.93 | |
| **Hospital bed days** | | | |
| Primary hospital | | 1.21 | WHO Choice [59] inflated to 2021 |
| Secondary hospital | | 1.26 | |
| Tertiary hospital | | 1.63 | |
| **Health facility visit** | | **0.00** | |
| Primary hospital | | 0.43 | |
| Secondary hospital | | 0.49 | |
| Tertiary hospital | | 0.51 | |
| Health center visit | | 0.53 | |
| PCI intervention | | 1448.28 | |
| In-patient costs for MI | | 1040.00 | |
| In-patient costs for Stroke | | 940.00 | |
| Outpatient cost for IHD (per annum) | | 45.00 | |
| Outpatient cost for Stroke (per annum) | | 67.00 | |
| **Salary scale of human resource** | | **0.00** | |
| Physician | | 485.06 | Federal Ministry of Health, Ethiopia 2012/2019 |
| Acute care nurse | | 171.72 | |
| Pharmacy personnel | | 184.99 | |
| Laboratory technician | | 148.51 | |
| **Program costs in 2021 $US** | | **Cost** | **Source** |
| Monitoring and Evaluation | | 18,760,664.35 | National strategic action plan for prevention & control of NCDs 2014–2016 [60]. |
| Health Promotion | | 11,435,622.57 | |
| National Systems Response | | 3,962,354.33 | |
| **Total national cost** | | 34,158,641.24 | |
| Program costs for our study area | | 1,024,759.24 | |
| Program cost per person per annum | | 128.28 | |
| Monthly program cost per person | | 10.69 | |
| Monthly program cost of study population | | 4,340.032 | |

1USD = 20.999 ETB in 2016 and 43.5 in 2021; PPP = 12.1/8.1 = 1.5.

MD: metered Dose; 1 USD = 43.5 January 2021.

**Note**: 30% mark-up at regional EPSA hub, 31% mark-up at Public Hospital level.

Estimated hypertensive population by using 21.39 mean prevalence of hypertension (+15 years) in southern Ethiopia = 750,533.

**Table 2. Treatment modalities for managing high blood pressure based on ISH 2020 guidelines among adult hypertensive patients in antihypertensive drug public hospitals in Southern Ethiopia (n = 406).**

| Classification of hypertension ISH 2020 | BP level | Target BP (optimal) | Anti-hypertensive regimen |
|---|---|---|---|
| Grade 1 hypertension | 140-159/90-99 mmHg | < 130/80 mmHg if tolerated in age < 65 years **OR** <140/90mmHg in ≥ 65 years | **Monotherapy** |
| | | | Amlodipine 5mg |
| | | | Nifedipine 20mg |
| | | | Enalapril 5mg |
| | | | HCT 25mg |
| | | | **ACEI + CCB = dual low dose combination** |
| | | | Amlodipine 5mg + Enalapril 5mg |
| | | | Nifedipine 20mg + Enalapril 5mg |
| | | | Amlodipine + Captopril |
| Grade 2 hypertension | ≥ 160/100 mmHg | | Nifedipine + Captopril |
| | | | **ACEI + CCB = dual full dose** |
| | | | Amlodipine + Enalapril |
| | | | Nifedipine + Enalapril |
| | | | Amlodipine + Captopril |
| Resistant hypertension | Hypertension despite treatment with 3 full standard dose first line antihypertensives | | Nifedipine + Captopril |
| | | | **ACEI + Diuretic = dual low dose** |
| | | | HCT + Enalapril |
| | | | HCT + Captopril |
| | | | **ACEI + Diuretic = dual full dose** |
| | | | HCT + Enalapril |
| | | | HCT + Captopril |
| | | | **Triple combination = ACEI + CCB + Diuretics** |
| | | | ACEI + CCB + Diuretics + Spironolactone 12.5 – 50mg |
| Acute MI | Patient stabilization and recovery | | ASA 325mg + Enalapril 20mg + Atenolol 50mg + Clpidogrel 300mg + Streptokinase 1.5 MIU |
| | | | Insertion of balloon tipped catheter with stent to blocked area (PCI) |
| Post-acute MI | Secondary prevention | | ASA 81mg + Enalapril 20mg + Atenolol 50mg + Simvastatin 40mg daily for 30 Days |
| Acute stroke | Patient stabilization and recovery | | ASA 160mg daily for 1 month |
| Post-acute stroke | Secondary prevention | | ASA 81mg + Enalapril 20mg + simvastatin 40mg daily for 30 days |
| Primary prevention of IHD and stroke | Primary prevention | | **HCT 25mg + Atenolol 50mg Daily + Simvastatin 40mg Daily** |
| Combination drugs for absolute CVD risk | Secondary prevention | | ASA 100mg + HCT 25mg + Atenolol 50mg + Simivastin 20mg |

## 3.2. Cost-effectiveness of treating hypertension

The average cost-effectiveness of treating hypertension based on the ISH 2020 guideline was 800.23 USD/DALY averted (i.e. 46, 268.92 USD per 57.82 DALYs averted). The incremental net monetary benefit of treating hypertension based on ISH 2020 guidelines was 128,520,077.61 USD by considering a willingness-to-pay threshold of $ 50,000 US per DALY averted. The incremental cost-effective ratio of treating hypertension when compared with null was 1,125.44 $US per DALY averted. This was cost-effective based on one to three times the GDP threshold of Ethiopia (936 to 2,808 $US in 2020) [79].

**Table 3. Projected annual productive loss due to premature mortality (YLL) due to hypertension and hypertension-related morbidity (YLD) in Southern Ethiopia.**

| Years lost due to premature morality | Years | YLL (ETB) | YLL ($US) |
|---|---|---|---|
| Male | 10,121 | 231,670,739.35 | $ 5,325,764.12 |
| Female | 9,737 | 141,710,242.32 | $ 3,257,706.72 |
| Both sexes | 19,858 | 373,380,981.67 | $ 8,603,248.43 |
| **Years lost due to hypertension morbidity** | **Years** | **YLD (ETB)** | **YLD ($US)** |
| Male | 717.86 | 16,431,889.83 | $ 377,744.59 |
| Female | 1,772.80 | 25,800,956.93 | $ 593,125.45 |
| Both sexes | 2,490.66 | 42,232,846.75 | $ 970,870.04 |
| Total projected annual productive life years loss | 22,348.66 | 415,613,828.42 | $ 9,574,118.47 |

**Note**: Formula for calculating YLL or YLD: For male: = ((YLL/YLD*2059.078*0.88)*12) + ((YLL/YLD*796*0.12)*12) and for female: = ((YLL/YLD*2059.078*0.33)*12) + ((YLL/YLD*796*0.67)*12).
1$US = 43.5 ETB.

We also evaluated the cost-effectiveness of treating hypertension at different ages and sex groups. Treating hypertension among adults aged 40–64 years was very cost-effective 625.27 USD per DALY averted (i.e., below 1 GDP per capita per DALY averted). Similarly, treating hypertension among adults aged 40–64 years was very cost-effective in men with ICER of 503.34 USD per DALY averted and in women with an Incremental cost-effectiveness ratio of 906.63 USD per DALY averted. However, treating hypertension below 40 years was not cost-effective 3,586.02 USD per DALY averted (i.e., above 2,483.75 USD, which is 3 times GDP per capita value) (Table 4).

## 3.3. Hypertension patients with DM and or CKD

Concerning the value for money in treating hypertension with diabetes and or chronic kidney disease (CKD), treating adult patients with hypertension and diabetes and or CKD could save 20,021,734.10 ETB (460,269.75 USD) annually. Treating hypertension among adults aged 40–64 years with diabetes, and or CKD was cost-effective with an ICER of 798.59 USD per DALY

**Table 4. Projected average annual incremental results of treating adults with untreated hypertension between the ages of 33 and 64 years (2020–2030).**

| Age category | Cost treated | Cost untreated | DALYs treated | DALYs untreated | Incremental cost | Incremental DALYs | ICER* | Remark |
|---|---|---|---|---|---|---|---|---|
| ≥ 65 years | 20,313.65 | 27,944.78 | 29.24 | 40.38 | 7,631.13 | 11.14 | 685.02 | 🟩 |
| 40–64 years | 688,568.54 | 1,062,069.53 | 922.48 | 1,519.82 | 373,500.99 | 597.34 | 625.27 | 🟩 |
| < 40 years | 243,259.23 | 416,786.63 | 103.36 | 151.75 | 173,527.40 | 48.39 | 3,585.02 | 🟥 |

**Hypertension treatment effectiveness by age and sex category**

| | Cost treated | | Cost untreated | | DALYs Treated | | DALYs Untreated | | **ICER Male** | ICER Female | Remark |
|---|---|---|---|---|---|---|---|---|---|---|---|
| | Male | Female | Male | Female | Male | Female | Male | Female | | | |
| ≥ 65 years | 8,902.42 | 11,411.22 | 8,732.35 | 19,212.43 | 351.05 | 796.99 | 9.29 | 31.09 | 0.49763 | -10.19 | 🟥 |
| 40–64 years | 260,567.37 | 428,001.17 | 328,634.76 | 733,434.76 | 311.19 | 733.50 | 446.42 | 1,070.39 | 503.34 | 906.63 | 🟩 |
| < 40 years | 213,460.15 | 29,799.08 | 359,549.53 | 57,237.10 | 39.89 | 63.49 | 58.53 | 93.22 | 7838.42 | 922.98 | 🟩 |

**Note**: GDP per capita in 2020 was 936.30 USD: The intervention is said to be cost-effective if it costs one to three times GDP per capita (936.30 to 2,808.90) per DALY averted.
*: USD/DALYs averted: DALYs: Disability-adjusted Life years

| 🟩 | **Very cost-effective**: Less than one GDP per capita per DALY averted (<936.30 USD) |
|---|---|
| 🟨 | **Cost-effective**: One to three GDP per capita per DALY averted (936.30 to 2,808.90 USD) |
| 🟥 | **Not cost-effective**: More than three GDP per capita per DALY averted (>2,808.90 USD) |

**Table 5. Projected average annual incremental results of treating adults with untreated hypertension and diabetes, and or chronic kidney diseases above the Ages of 33 Years (2020–2030).**

| Age category | Cost treated | | Cost untreated | | DALYs treated | | DALYs untreated | | ICER* | | Remarks |
|---|---|---|---|---|---|---|---|---|---|---|---|
| ≥ 65 years | 2,603.17 | | 2,851.43 | | 2.62 | | 3.84 | | 203.49 | | 🟩 |
| 40–64 years | 581,694.25 | | 902,913.17 | | 1,149.12 | | 1,551.36 | | 798.58 | | 🟩 |
| < 40 years | 152,971.26 | | 291,773.84 | | 82.50 | | 121.13 | | 3,593.13 | | 🟥 |

**Treatment effectiveness by age and sex category patients with diabetes, and or CKD**

| Age category | Cost treated | | Cost untreated | | DALYs Treated | | DALYs untreated | | ICER Male | ICER Female | Remarks |
|---|---|---|---|---|---|---|---|---|---|---|---|
| | Male | Female | Male | Female | Male | Female | Male | Female | | | |
| 40–64 years | 227,818.67 | 353,875.57 | 298,226.64 | 604,686.53 | 364.25 | 784.24 | 490.09 | 1061.26 | 559.48 | 905.40 | 🟩 |
| < 40 years | 126,500.45 | 26,470.81 | 240,958.59 | 50,815.24 | 26.14 | 35.36 | 38.37 | 82.753 | 9353.91 | 513.72 | 🟩 |
| ≥ 65 years | 2,603.17 | 0.00 | 2,851.42 | 0.00 | 2.17 | 0 | 3.84 | 0 | 148.15 | - | 🟥 |

| | |
|---|---|
| 🟩 | **Very cost-effective**: Less than one GDP per capita per DALY averted (<936.30 USD) |
| 🟨 | **Cost-effective**: One to three GDP per capita per DALY averted (936.30 to 2,808.90 USD) |
| 🟥 | **Not cost-effective**: More than three GDP per capita per DALY averted (>2,808.90 USD) |

**Note**: Projected Average Cost-Effectiveness of Full Implementation of the 2020 ISH Guidelines for hypertension treatment in patients without Cardiovascular Disease, According to Sex, Age, Hypertension Stage, and Status concerning Diabetes and Chronic Kidney Disease.

CKD; chronic kidney disease, DALYs; disability-adjusted life years, and ICER; incremental cost-effectiveness ratio.

* USD/DALYs averted

averted. Treating adults below 40 years with hypertension and diabetes and or CKD was cost-effective in women with ICER of 513.72 USD per DALY averted but not cost-effective in men with ICER of 9,353.91USD per DALY averted. Treating hypertensive adults aged 40–64 years with diabetes and CKD is very cost-effective (i.e., less than 1 GDP per capita per DALY averted) in both women and men (i.e., 559.48 USD and 905.40 USD/DALY averted respectively). We also evaluated the cost-effectiveness of treating hypertension based on a half per capita income threshold value. Based on this threshold (468.15 USD) treating hypertension at all ages for both sexes is not cost-effective. However, treating hypertension patients with diabetes and or CKD aged 65 years and above is cost-effective based on half per capita income threshold with an ICER of 202.54 USD per DALY averted) (Table 5).

## 3.4. Hypertension in adults without DM and or CKD

Treating hypertension in adults without diabetes or CKD was not cost-effective below 40 years with an ICER value of 3557.872 USD per DALY averted. Treating hypertension in patients 40 years and above without diabetes and or CKD was cost-effective for both men and women with ICER of 1,101.59 USD per DALY averted in men and 1,034.52 USD per DALY averted in women (Table 6).

## 3.5. Sensitivity analyses

Deterministic sensitivity analyses were done to assess the impact of uncertainty around key data inputs on the model outputs by using lower and upper estimates (S1–S5 Figs). These included the prevalence of hypertension (19.6 to 41.9%), discounting rate of 6%, cost of antihypertensive ± 20% variation, and blood pressure control rate of 50 to 70% [22, 35]. In addition, we modeled adherence levels that were 50% and 75% lower than those in clinical trials, penalties for pill-taking disutility 0.049 (0.031–0.072) (i.e., a decrease in the quality of life associated with taking a medication) [6], and a 1-year lag in achieving the target blood pressure. We also used probabilistic (Monte Carlo) simulation to sample uncertainty distributions concerning

**Table 6. Projected average annual incremental results of treating adults with untreated hypertension without diabetes, and or chronic kidney diseases between the ages of 33 and 64 Years (2020–2030).**

| Age category | Cost treated | | Cost untreated | | DALYs treated | DALYs untreated | ICER* | Remarks |
|---|---|---|---|---|---|---|---|---|
| 40–64 | 105,770.27 | | 168,181.09 | | 214.60 | 291.76 | 808.8494 | 🟩 |
| ≥ 65 years | 24,069.13 | | 29,865.03 | | 31.26 | 41.11 | 588.4171 | 🟩 |
| < 40 years | 90,287.98 | | 125,012.80 | | 20.86 | 30.62 | 3557.872 | 🟥 |

**Treatment effectiveness by age and sex category patients with no diabetes, and or CKD**

| Age category | Cost treated | | Cost untreated | | DALYs Treated | | DALYs Untreated | | ICER Male | ICER Female | Remarks |
|---|---|---|---|---|---|---|---|---|---|---|---|
| | Male | Female | Male | Female | Male | Female | Male | Female | | | |
| 40–64 years | 31,644.69 | 74,125.60 | 39,432.90 | 128,748.22 | -53.06 | -50.74 | -43.67 | 9.13 | 1,101.59 | 1,034.52 | 🟨 |
| < 40 years | 86,959.70 | 3,328.27 | 118,590.94 | 6,421.86 | 37.35 | 28.13 | 20.16 | 10.47 | -3,968.79 | -319.16 | 🟥 |
| ≥ 65 years | 8,830.76 | 15,238.36 | 7,568.97 | 22,296.07 | 348.88 | 0 | 5.45 | 0.00 | -231.52 | -1,294.99 | 🟥 |

| 🟩 | **Very cost-effective**: Less than one GDP per capita per DALY averted (<936.30 USD) |
|---|---|
| 🟨 | **Cost-effective**: One to three GDP per capita per DALY averted (936.30 to 2,808.90 USD) |
| 🟥 | **Not cost-effective**: More than three GDP per capita per DALY averted (>2,808.90 USD) |

CKD; chronic kidney disease, DALYs; disability-adjusted life years, and ICER; incremental cost-effectiveness ratio.

* USD/DALYs averted; 1USD = 43.5 ETB.

the effectiveness of blood pressure lowering with the use of antihypertensive drugs, the relative risk reduction in CVD with treatment, disutility penalty for the daily taking of antihypertensives, and side effects of medications. Uncertainty distributions were randomly sampled 1000 times, and 95% uncertainty intervals were calculated. Almost all probabilistic sensitivity analyses (≥99.7% of simulation results) predicted cost savings for the treatment of patients with hypertension, except for those in the age group below 40 years. The results were robust to all included variables. (S5 Fig).

## 4. Discussion

In this study, we evaluated the cost-effectiveness of treating hypertension based on ISH 2020 guidelines by using a modified CVD policy adapted for the Sub-Saharan African context [9–11, 15]. Treatment hypertension to ISH 2020 guidelines target BP with a 75% control rate and was projected to prevent approximately 22,348.66 total productive life-year losses annually. This equates 9,574,118.47 $US in 2021. This is higher than 11,050 reported deaths (i.e. 30 patients per day) caused by hypertension in 2017 [88]. This could be explained by the increasing trend of hypertension and inadequate BP control in the country including in our study area. A cost-effectiveness analysis conducted to evaluate the cost-effectiveness of prevention and treatment of cardiovascular disease in Ethiopia showed that combination drug treatment for individuals having >35% absolute risk of a CVD event in the next 10 years is cost-effective with ICER of US$67 per DALY averted and about US $7 million annually [89]. This is also supported by evidence from a study conducted to evaluate the cost-effectiveness of hypertension therapy according to 2014 guidelines by using cardiovascular disease policy model-based simulation modeling in the USA showed that full implementation of the new hypertension guidelines would result in approximately 56,000 fewer cardiovascular events and 13,000 fewer deaths 13,000 deaths annually [90].

The average cost-effectiveness of treating hypertension was 800.23 USD/DALY averted (i.e. 46, 268.92 USD per 57.82 DALYs averted). The incremental net monetary benefit of treating hypertension was 128,520,077.61 USD by considering a willingness-to-pay threshold of $ 50,000 US per DALY averted and 20,151,503 USD by considering a willingness-to-pay threshold of $ 2,808.90 US per DALY averted (three times National GDP per capita).

A modeling study conducted to evaluate the cost-effectiveness of improved hypertension management in India showed that at 70% coverage and adherence, the hypertension control intervention would be cost-saving overall [91].

The incremental cost-effective ratio of treating hypertension when compared with null was 1,125.44 $US per DALY averted. This was cost-effective based on one to three times the GDP threshold of Ethiopia (936 to 2,808 $US in 2020) [79]. A WHO-CHOICE analysis of the cost-effectiveness of population-level and individual-level interventions to combat non-communicable disease in Eastern sub-Saharan Africa Asia showed that treatment options for cardiovascular disease (CVD) are highly cost-effective (ICER < 100 in.$ per DALY averted) [92].

In this study, treating hypertension among adults aged 40–64 years was very cost-effective with an ICER of 625.27 USD per DALY averted. Treating hypertensive adults aged 40–64 years was more cost-effective for men than women with ICER of, 503.34 USD per DALY averted 906.63 USD per DALY averted respectively. This could be due to the high risk of CVD risk factors like smoking, and alcohol consumption in men than in women. Because at a younger age, coronary heart disease (CHD) incidence and mortality were ≈3-fold and ≈5-fold greater in men. The differences in smoking rate contributed markedly to the excess CHD risk of men because smoking could also decrease HDL cholesterol levels [93].

This is supported by evidence from a study conducted to evaluate the cost-effectiveness of hypertension therapy according to 2014 guidelines in the USA showed that treatment of stage 1 hypertension was intermediate or low cost-effective for women between the ages of 35 and 44 years [90].

Treating adults with hypertension and diabetes and or CKD could save 460,269.75 USD annually. Treating hypertension among adults aged above 40 years with diabetes, and or CKD was cost-effective with ICER of 798.59 USD per DALY averted for age 40–64 years and 203.49 USD per DALY averted for age 65 years and above. Treating hypertension in adults without diabetes or CKD was not cost-effective below 40 years with an ICER of 3,593.13 USD per DALY averted. This is in line with the existing evidence available on clinical practice guidelines that suggested treating high-risk patients for reducing cardiovascular and cerebrovascular complications and all-cause mortality among hypertensive populations [94–96]. Treating hypertensive adults aged 40–64 years with diabetes and CKD is very cost-effective for men and cost-effective for women. This could be explained by variation in risk of cardiovascular and cerebrovascular complications, and life expectancy in men and women.

Treating hypertension in patients without diabetes and or CKD was cost-effective for both men and women aged 40–64 years with ICER of 1,101.59 USD per DALYs averted in men and 1,034.52 USD per DALY averted in women. This is supported by a simulated modeling study conducted for a 10-year time horizon to evaluate the cost-effectiveness of hypertension treatment in the USA showed that for stage 1 HTN but without diabetes or CKD, cost savings extended to ICER $57,000/QALY & $46,000/QALY in non-Hispanic black males and females aged 35–44 years respectively [16].

## 5. Strengths and limitations

We used the CVD policy model adapted for the Sub-Saharan African perspective. The results were reported based on Consolidated Health Economic Evaluation Reporting Standards (CHEERS). All transition probabilities of cardiovascular and stroke events and effectiveness assumptions were based on a systematic review and meta-analysis of randomized trials. Valuation of productivity loss was done from a societal perspective. One way deterministic sensitivity analysis and probabilistic sensitivity analysis were done. However, we could underestimate the national economic loss associated with hypertension due to, the use of computer

simulation model-based analysis with several assumptions, uncertainty in age, and sex-specific prevalence of undiagnosed hypertension, using data from multiple sourcing. In addition to this, we did not analyze the effect of a healthy lifestyle on lowering blood pressure and specific antihypertensive medicines. Finally, we did not consider the role of implementing different guideline recommendations.

## 6. Conclusion

In conclusion, implementation of the 2020 ISH guidelines for Ethiopian hypertensive adults between the ages of 33 and 64 years could potentially prevent about 13,865 total deaths annually, and 46,501,793.85 USD economic losses. Treating hypertension without diabetes and or CKD below 40 years is not cost-effective based on one to three times the GDP per capita threshold value. Treating hypertension for patients 40–64 years old is cost-effective. Therefore, improving treatment coverage, blood pressure control rate, and adherence to treatment by involving all relevant stakeholders at all levels is critical to saving scarce health resources.

## Supporting information

**S1 Fig. Proposed one cycle Model flow diagram for cost of hypertension in Ethiopia.**
(TIF)

**S2 Fig. Cost-effectiveness acceptability curve based on the lower and upper bound of disutility for daily taking of antihypertensive medications.**
(TIF)

**S3 Fig. Cost-effectiveness acceptability curve based on ±20% change of cost hypertension treatment.**
(TIF)

**S4 Fig. Cost-effectiveness acceptability curve based on the lower bound discounting costs at 6% rate.**
(TIF)

**S5 Fig. ICER scatter plot and cost-effectiveness acceptability curve based on probabilistic sensitivity analysis.**
(TIF)

**S1 Table. Model parameters and probability of transition between states.**
(DOCX)

**S2 Table. Simulation input parameters and disability weights for hypertension and related complications the global burden of disease 2013 study and WHO Global Health Estimates.**
(DOCX)

**S3 Table. World Health Organization (WHO); Ethiopian life table 2019.**
(DOCX)

**S4 Table. Estimated working age population and percent distribution of women and men age 15–64 by employment status, according to, Ethiopia DHS 2016 and national STEPS survey and world population prospect, 2020.**
(DOCX)

**S5 Table. Ethiopian population 2020 estimate and prevalence of hypertension in Ethiopia.**
(DOCX)

**S6 Table. Risk of death across age and gender covariate categories stratified for hypertension.**
(DOCX)

**S7 Table. Age-sex specific mortality.**
(DOCX)

**S8 Table. Percent distribution of adult mortality rates, among 15–49, Ethiopia DHS 2016.**
(DOCX)

**S9 Table. Annual mortality rate in the total population, those with hypertension by treatment and control status and those without hypertension in Ethiopia in 2021 by age group and sex based on literature review of systematic reviews and clinical trials.**
(DOCX)

**S10 Table. Estimates of preventive effect of taking one or more blood pressure lowering drugs on coronary heart disease (CHD) events and stroke according to pretreatment systolic blood pressure, age, number of drugs, and dose (as multiple of standard).**
(DOCX)

**S11 Table. Estimates of preventive effect of taking one or more blood pressure lowering drugs on coronary heart disease (CHD) events and stroke according to pretreatment diastolic blood pressure, age, number of drugs, and dose (as multiple of standard33)32 then the effect of this blood pressure reduction on disease risk.**
(DOCX)

**S1 File. Survey questionnaire.**
(PDF)

**S2 File.**
(DOCX)

**S1 Data. Markov model data for CEA hypertension.**
(XLTM)

## Author Contributions

**Conceptualization:** Majid Davari, Mende Mensa Sorato, Abbas Kebriaeezadeh, Nizal Sarrafzadegan.

**Data curation:** Mende Mensa Sorato.

**Formal analysis:** Mende Mensa Sorato.

**Methodology:** Majid Davari, Mende Mensa Sorato, Abbas Kebriaeezadeh, Nizal Sarrafzadegan.

**Writing – original draft:** Mende Mensa Sorato.

**Writing – review & editing:** Majid Davari, Mende Mensa Sorato, Abbas Kebriaeezadeh, Nizal Sarrafzadegan.

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
