## [Decision Letter · Decision Letter 0]

30 May 2022

PONE-D-21-28063Cost-effectiveness of hypertension therapy based on 2020 International society of hypertension guidelines in Ethiopia from a societal perspectivePLOS ONE

Dear Dr. Sorato,

Thank you for submitting your manuscript to PLOS ONE. After careful consideration, we feel that it has merit but does not fully meet PLOS ONE’s publication criteria as it currently stands. Therefore, we invite you to submit a revised version of the manuscript that addresses the points raised during the review process.

We look forward to receiving your revised manuscript.

Kind regards,

Vanessa Carels

Staff Editor

PLOS ONE

Journal Requirements:

Reviewers' comments:

Reviewer's Responses to Questions

**Comments to the Author**

1. Is the manuscript technically sound, and do the data support the conclusions?

Reviewer #1: Yes

2. Has the statistical analysis been performed appropriately and rigorously? 

Reviewer #1: Yes

3. Have the authors made all data underlying the findings in their manuscript fully available?

Reviewer #1: Yes

4. Is the manuscript presented in an intelligible fashion and written in standard English?

Reviewer #1: No

5. Review Comments to the Author

Reviewer #1: General comments: the manuscript requires a major English grammar revision, but the study presented is robust and relevant.

Abstract:

“2020 international society of hypertension (ISH)” – capital letters for the name of the society (2020 International Society of Hypertension). In many other parts of the text the same problem has been identified.

The acronym ICER is not explained when it first appears.

Introduction

These sentences are not clear. They should be rewritten as part of a separate paragraph and would improve from adding some examples of its use, especially in LMIC, and its general rationale.

“The cardiovascular disease policy model which addresses all relevant contributors to the model is used by many hypertension cost-effectiveness studies. It is a computer-simulation, state-transition (Markov cohort) model of coronary heart disease and stroke incidence, prevalence, mortality, and costs over age 35 years (9-11).”

Methods

In section 2.2 (Study design) part of the information on the model could be transferred to the introduction. The modeling rationale and details are well presented.

Nevertheless, for the productivity costs attributable to hypertension, why were YLL (years of life lost) used instead of YPLL (years of productive life lost) as normally applied for Human Capital Approach methods? That means considering the years until retirement and not to full life expectancy. Additionally, were any discount rates applied to the estimates?

Regarding the sensitivity analyses, why were probabilistic simulations (Monte Carlo) limited to 1,000 draws? In general, final results of modelling studies are presented for 5,000 or 10,000 Monte Carlo draws for purposes of the robustness of the estimates.

Results

The study is based on estimates of the projected impact of hypertension treatment. Commonly recent modelling studies use rounded estimates (ie. To the nearest thousand or using to 2 or 3 significant digits), so that a false idea of precision is avoided in the results. Additionally, it would be helpful to add the uncertainty intervals to the tables and overall results.

Discussion

In terms of the limitations, it seems as no time-lag between the implementation of the guidelines and changes in outcomes was considered in the modeling.

6. PLOS authors have the option to publish the peer review history of their article (what does this mean?). If published, this will include your full peer review and any attached files.

Reviewer #1: No

---

## [Author Response · Author response to Decision Letter 0]

7 Jun 2022

Title: Cost-effectiveness of hypertension therapy based on 2020 International society of hypertension guidelines in Ethiopia from a societal perspective

Thank you for showing gaps in our manuscript. I accepted some your suggestions. I answered questions which require clarification as follows. 

Reviewer #1: General comments: the manuscript requires a major English grammar revision, but the study presented is robust and relevant.

Answer: English re-edited 

Abstract:

“2020 international society of hypertension (ISH)” – capital letters for the name of the society (2020 International Society of Hypertension). In many other parts of the text the same problem has been identified.

The acronym ICER is not explained when it first appears.

Answer: Modified 

Introduction

These sentences are not clear. They should be rewritten as part of a separate paragraph and would improve from adding some examples of its use, especially in LMIC, and its general rationale.

“The cardiovascular disease policy model which addresses all relevant contributors to the model is used by many hypertension cost-effectiveness studies. It is a computer-simulation, state-transition (Markov cohort) model of coronary heart disease and stroke incidence, prevalence, mortality, and costs over age 35 years (9-11).”

Answer: Modified

Methods

In section 2.2 (Study design) part of the information on the model could be transferred to the introduction. The modeling rationale and details are well presented.

Nevertheless, for the productivity costs attributable to hypertension, why were YLL (years of life lost) used instead of YPLL (years of productive life lost) as normally applied for Human Capital Approach methods? That means considering the years until retirement and not to full life expectancy. Additionally, were any discount rates applied to the estimates?

Answer: We used YLL, since we have Value of lost productivity questionnaire that excludes population above 64 years. We assigned zero for productivity loss for people above 64 years. This is equivalent with YPLL. It is provided as supplementary material entitled “markov-model for Cost-effectiveness analysis of hypertension Excel file”. 

We adjusted for presence of comorbidities during estimation of YLL or YPLL. According to the GBD, 2010 study simplified calculation of DALYs was recommended. Therefore, we used a new normative standard life table for the loss function used to compute YLLs; calculate of YLDs simply as the prevalence of each sequela multiplied by the relevant disability weight; adjust for comorbidity in the calculation of YLDs, and no discounting for time or unequal age weights (76).

Regarding the sensitivity analyses, why were probabilistic simulations (Monte Carlo) limited to 1,000 draws? In general, final results of modelling studies are presented for 5,000 or 10,000 Monte Carlo draws for purposes of the robustness of the estimates.

Answer: Monte Carlo Simulation with 1,000 Iterations is minimum requirement for checking robustness of estimates. We used the minimum iteration size. 

Results

The study is based on estimates of the projected impact of hypertension treatment. Commonly recent modelling studies use rounded estimates (i.e. To the nearest thousand or using to 2 or 3 significant digits), so that a false idea of precision is avoided in the results. Additionally, it would be helpful to add the uncertainty intervals to the tables and overall results.

Answer. We made many improvements on result section after revising the analysis many times. 

Discussion

In terms of the limitations, it seems as no time-lag between the implementation of the guidelines and changes in outcomes was considered in the modeling.

Answer: The analysis was conducted on year 2021 and mentioning time lag as study limitation has little importance. We listed limitations that could affect the outcomes only.

---

## [Editor Report · Decision Letter 1]

9 Aug 2022

Cost-effectiveness of hypertension therapy based on 2020 International Society of Hypertension guidelines in Ethiopia from a societal perspective

PONE-D-21-28063R1

Dear Dr. Mende Mensa Sorato

We’re pleased to inform you that your manuscript has been judged scientifically suitable for publication and will be formally accepted for publication once it meets all outstanding technical requirements.

Kind regards,

Eduardo Augusto Fernandes Nilson

Guest Editor

PLOS ONE

Additional Editor Comments (optional):

Thank you for addressing all the points from the peer reviewers.
---

## [Editor Report · Acceptance letter]

12 Aug 2022

PONE-D-21-28063R1 

Cost-effectiveness of hypertension therapy based on 2020 International Society of Hypertension guidelines in Ethiopia from a societal perspective 

Dear Dr. sorato:

I'm pleased to inform you that your manuscript has been deemed suitable for publication in PLOS ONE. Congratulations! Your manuscript is now with our production department. 

Kind regards, 

on behalf of

Dr. Eduardo Augusto Fernandes Nilson 

Guest Editor

PLOS ONE